# Fluorometric Detection of Thiamine Based on Hemoglobin–Cu_3_(PO_4_)_2_ Nanoflowers (NFs) with Peroxidase Mimetic Activity

**DOI:** 10.3390/s20216359

**Published:** 2020-11-07

**Authors:** Hangjin Zou, Yang Zhang, Chuhan Zhang, Rongtian Sheng, Xinming Zhang, Yanfei Qi

**Affiliations:** School of Public Health, Jilin University, Changchun 130021, Jilin, China; zouhj18@mails.jlu.edu.cn (H.Z.); yangzhang19@mails.jlu.edu.cn (Y.Z.); zhangch18@mails.jlu.edu.cn (C.Z.); shengrt19@mails.jlu.edu.cn (R.S.); xmzhang19@mails.jlu.edu.cn (X.Z.)

**Keywords:** food analysis, fluorometric method, thiamine, nanozymes, protein-inorganic nanoflowers

## Abstract

Component analysis plays an important role in food production, pharmaceutics and agriculture. Nanozymes have attracted wide attention in analytical applications for their enzyme-like properties. In this work, a fluorometric method is described for the determination of thiamine (TH) (vitamin B_1_) based on hemoglobin–Cu_3_(PO_4_)_2_ nanoflowers (Hb–Cu_3_(PO_4_)_2_ NFs) with peroxidase-like properties. The Hb–Cu_3_(PO_4_)_2_ NFs catalyzed the decomposition of H_2_O_2_ into ·OH radicals in an alkaline solution that could efficiently react with nonfluorescent thiamine to fluoresce thiochrome. The fluorescence of thiochrome was further enhanced with a nonionic surfactant, Tween 80. Under optimal reaction conditions, the linear range for thiamine was from 5 × 10^−8^ to 5 × 10^−5^ mol/L. The correlation coefficient for the calibration curve and the limit of detection (LOD) were 0.9972 and 4.8 × 10^−8^ mol/L, respectively. The other vitamins did not bring about any obvious changes in fluorescence. The developed method based on hybrid nanoflowers is specific, pragmatically simple and sensitive, and has potential for application in thiamine detection.

## 1. Introduction

Thiamine (TH) (vitamin B_1_) is an important water-soluble vitamin in the human body [1,2]. It is involved in various metabolic processes [3], and the phosphorylated derivative thiamine diphosphate is a transketolase for the substances necessary for the activity of mitochondrial pyruvate [4]. Humans and other mammals cannot synthesize thiamine by themselves, and instead, take in vitamin B_1_ from the outside world [5]. The recommended daily intake of vitamin B_1_ in adults is 0.5–1 mg [6]. A severe deficiency of vitamin B_1_ can cause athlete’s foot, neuralgia, optic neuropathy, anorexia and KORSAKOFF syndrome [7,8,9]. Between 3 and 5 mg/kg of vitamin B_1_ are required as a food additive in rice, wheat and miscellaneous grains. The related detection methods for vitamin B_1_ involve spectrophotometry [10,11], chemiluminescence [12], chromatography [13,14,15], fluorescence [7,16,17], electrochemistry [18] and capillary electrophoresis [19]. Notably, although some of them are capable of accurate or high-throughput monitoring of amounts of vitamin B_1_ and have even been practically used in food quality control, these methods are restricted to expensive, low selectivity or complicated experimental conditions. Among these methods, the fluorometric method based on the oxidation of thiamine to thiochrome by potassium hexacyanoferrate (III) in alkaline solution is the most widely used. However, the thiochrome yield (about 67%) of the reaction using hexacyanoferrate as an oxidant is not high. Therefore, this method is often improved by substituting hexacyanoferrate with other materials, such as graphene quantum dot-capped gold nanoparticles [20] and oxygen vacancy-engineered PEGylated MoO_3-x_ nanoparticles [21]. These materials offer improved sensitivity and assay simplicity with unexpected properties including large surface-to-volume ratio, high stability and enhanced oxidizing capability.

Since Zare et al. [22] first reported protein-inorganic nanoflowers, a variety of new protein-inorganic hybrid nanoflowers have been synthesized with greater specific surface area, better stability and cooperatively enhanced catalysis features. The composite materials have shown potential for application in sensors and analytical devices and in biocatalysis. Qu et al. further explored the peroxidase-like activity of protein-inorganic hybrid nanoflowers and performed self-activated cascade reactions [23]. Recently, Pang et al. reported Hb–Cu_3_(PO_4_)_2_ nanoflowers (NFs) with hemoglobin and Cu_3_(PO_4_)_2_ and enhanced peroxidase catalytic activity in the detection of hydrogen peroxide in an acetic acid buffer solution [24]. Yang et al. synthesized horseradish peroxidase-Cu_3_(PO_4_)_2_·3H_2_O hybrid nanoflowers for the fast and sensitive visual detection of H_2_O_2_ and phenol [25]. Despite this attractive feature of enzyme-Cu_3_(PO_4_)_2_·3H_2_O hybrid nanoflowers, the advantages and properties of hybrid nanoflowers have not yet been fully demonstrated. In this paper, we first investigate the enzyme-mimic activity of Hb–Cu_3_(PO_4_)_2_ NFs in an alkaline solution in detail. A Hb–Cu_3_(PO_4_)_2_ NF-based fluorescemetric method is developed for the detection of thiamine. Through a fluorometric assay, the alkaline solution, substrate, Hb–Cu_3_(PO_4_)_2_ NF concentrations and Tween 80 are investigated. Under optimal reaction conditions, the resulting sensor displayed a rapid response and high sensitivity to thiamine.

## 2. Materials and Methods

### 2.1. Materials and Chemicals

All the chemicals used were of analysis grade without further purification. Hemoglobin (Hb, from bovine blood) was purchased from Sigma-Aldrich. Thiamine, L-ascorbic acid (Vc) and folic acid (VB_9_) were obtained from Guoyao Chemical Research Institute (Shenyang, China). NaCl, KCl, NaHCO_3_, NaHPO_4_, NaH_2_PO_4_, NaHSO_4_, NaOH, glucose and hydrogen peroxide (H_2_O_2_, 30%) were purchased from Beijing Chemical Works (Beijing, China). 1,2-diaminobenzene (OPD) and CuSO_4_ were purchased from Tianjin Guangfu Fine Chemical Research Institute (Tianjin, China). The water used in the experiments was purified. The ρ of the water was 18 MΩ·cm.

### 2.2. Instrument

The fluorescence measurements were performed on a Shimadzu RF-5301 PC fluorophotometer (Kyoto, Japan) and Microplate reader (BioTek, Winooski, VT, USA). A 1 cm path length quartz cuvette and 96-well microplates were used in the experiments. The widths of the excitation and emission slits of the fluorophotometer were set to 3.0 and 3.0 nm, respectively. The Fourier-transform infrared (FTIR) spectrum was recorded in the range of 400–4000 cm^−1^ on KBr (FTIR IRAffinity-1s, Shimadzu, Japan). The pH measurements were performed by a PHS-25 pH meter (Shanghai INESA Scientific Instrument Co. Ltd., Shanghai, China). The UV–vis spectrum was recorded in the range of 200–1100 nm on a UV–vis spectrophotometer (Puxi Inc., Beijing, China). Zeta potential was measured by photon correlation spectroscopy using a Zetasizer (Nano-ZS90). A scanning electron microscopy (SEM) image was characterized by a Zeiss Merlin at 1.0 kV. The purified water was obtained from a SMART-N Heal Force Water Purification System (Shanghai Canrex Analytic Instrument Co., Ltd., Pudong, Shanghai, China). The nanoparticle surface charge was determined with a Malvern Zetasizer. The P element was determined by an inductively coupled plasma mass spectrometer (ELAN DRC-e, PerkinElmer, Concord, Ontario, Canada).

### 2.3. Synthesis of Hb–Cu_3_(PO_4_)_2_ NFs

Hb–Cu_3_(PO_4_)_2_ NFs were prepared according to a method described in the literature, with some modifications [24]. Briefly, 2 mL of CuSO_4_ aqueous solution (120 mM) was added into 300 mL of phosphate-buffered saline (PBS) solution (0.1 M, pH 7.4) containing 30 mg Hb. After incubation at 25 °C for 72 h, the turquoise precipitates of Hb–Cu_3_(PO_4_)_2_ NFs were collected by centrifugation (3000 rpm for 5 min) and washed with ultrapure water three times.

### 2.4. Detection of Reactive Hydroxyl Radical (·OH) Production

The hydroxyl radical (·OH) production was measured using a fluorescence method. Terephthalic acid (TA) was used as a fluorescence probe for detection of ·OH in H_2_O_2_. First, 75 μL of 25 mM TA in the NaOH (pH 13) solution was added into the 3 mL of PBS (pH 7.4) containing 100 mM H_2_O_2_ and 2 mg/mL Hb–Cu_3_(PO_4_)_2_ NFs. After 1 h incubation in the dark, the solution was examined for fluorometric measurements. Fluorescence spectra were obtained with an excitation wavelength of 315 nm and the emission spectra were recorded at a wavelength of 425 nm.

### 2.5. Enzyme-like Activity and Kinetic Parameter of Hb–Cu_3_(PO_4_)_2_ NFs

The enzyme-like activity of freshly synthesized Hb–Cu_3_(PO_4_)_2_ NFs was investigated spectrophotometrically by measuring the formation of DAP (2,3-diaminophenazine) from OPD at 450 nm (ε = 21,000 M^−1^ × cm^−1^) using a multiwell plate reader. Typically, 10 μL OPD (2 mM) was added into different 170 μL buffer solutions, followed by 10 μL Hb–Cu_3_(PO_4_)_2_ NFs (2 mg/mL) and 10 μL H_2_O_2_ (100 mM) at 37 °C. The activity of the Hb–Cu_3_(PO_4_)_2_ NFs at different pH values (pH 3–13) was studied using similar conditions as those described above. The pH of the different solutions was measured using a pH meter. Additionally, the buffers with pH values from 9 to 13 were prepared from the mixture of 0.2 M disodium hydrogen phosphate and a sufficient quantity of 1 M sodium hydroxide. pH values from 5 to 8 were obtained from the mixture of 0.2 M disodium hydrogen phosphate and 0.2 M sodium dihydrogen phosphate. Solutions with pH values of 3 and 4 were obtained from the mixture of 0.1 M citric acid and sodium citrate.

The steady-state kinetics of Hb–Cu_3_(PO_4_)_2_ NFs were performed by varying the concentrations of H_2_O_2_ (0–0.1 M) or OPD (0–2 mM) one at a time. The reaction was carried out in 160 μL phosphate buffers (pH 10) and monitored spectrophotometrically every 300 s using a multiwell plate reader. The kinetic curves were adjusted to the Michaelis–Menten model using the Origin (version 8.0, OriginLAB Corporation, Boston, MA, USA) software. The apparent kinetic parameters, Michaelis–Menten constant (K_m_) and maximum rate of reaction (V_max_) were calculated.

### 2.6. Condition Optimization

The reaction parameters of the time, pH, H_2_O_2_ and Hb–Cu_3_(PO_4_)_2_ NF concentrations as well as the fluorescence sensitizer (Tween 80) were investigated. Typically, 160 μL of different buffers (pH 8–13), 10 μL of different concentrations of Hb–Cu_3_(PO_4_)_2_ NFs (0–4 mg/mL), 10 μL of different concentrations of H_2_O_2_ (0–10 M) and 20 μL of 1 mM thiamine were mixed at room temperature. The fluorescence intensities were recorded every 15 min.

### 2.7. Fluorescent Detection of Thiamine

The quantitative determination of thiamine using a Hb–Cu_3_(PO_4_)_2_ NF-catalyzed fluorescent assay in the presence of H_2_O_2_ was performed as follows. A 160 μL Na_2_HPO_4_-NaOH buffer (pH 10) with 0.4% Tween 80, 10 μL of 100 mM H_2_O_2_, 20 μL of different concentrations of thiamine (0.0001–10 mM) and 10 μL of 2 mg/mL Hb–Cu_3_(PO_4_)_2_ NFs were mixed at room temperature, sequentially. Then, the mixtures were placed in a constant temperature incubator at 37 °C for 5 min. The fluorescence values of the above-mentioned solutions were measured using a microplate reader with an excitation wavelength of 370 nm. The emission spectra of these reaction solutions were recorded at a wavelength of 441 nm.

## 3. Results and Discussion

### 3.1. Characterization of Hb–Cu_3_(PO_4_)_2_ NFs

The Hb–Cu_3_(PO_4_)_2_ NFs were characterized using IR, UV–vis, SEM and Zetasizer methods. The IR spectra of CuSO_4_, hemoglobin and Hb–Cu_3_(PO_4_)_2_ NFs are shown in Figure 1a. The main characteristic vibration peaks of Hb were found at 1134, 1317 and 1679 cm^−1^ and attributed to v(C-N) and v(N-H), respectively. The peak found at 3306 cm^−1^ was assigned to the O-H of water. The spectrum of Hb–Cu_3_(PO_4_)_2_ NFs maintained the characteristic peaks of Hb at 1049, 1652 and 1734 cm^−1^ assigned to νas(N-H) and νas(C-N). The bands at 1041 and 1162 cm^−1^ were attributed to P-O and P=O vibrations. The existence of a phosphate element was also proved using an inductively coupled plasma mass spectrometer. The result was similar to that found in the literature [24]. As shown in Figure 1b, the UV–vis spectrum of Hb exhibited an absorbance at 254 nm. Cu^2+^ exhibited an absorbance at 800 nm. The characteristic peaks of Cu^2+^ and Hb were observed in the spectrum of the Hb–Cu_3_(PO_4_)_2_ NFs. The morphological structure of Hb–Cu_3_(PO_4_)_2_ NFs obtained by SEM is shown in Figure 1c. The SEM images indicate that Hb–Cu_3_(PO_4_)_2_ NFs have a flower-like structure with a size of about 15 μm. The surface zeta potential measurements show that the ζ of CuSO_4_, hemoglobin and Hb–Cu_3_(PO_4_)_2_ NFs were approximately −2.1 mV, 10.7 mV and −4.5 mV, respectively, as shown in Figure 1d. The ζ and the SEM consistently proved that the Hb–Cu_3_(PO_4_)_2_ NFs were successfully synthesized.

### 3.2. Enzyme-Like Activities and Kinetic Parameters of Hb–Cu_3_(PO_4_)_2_ NFs

In order to clarify the mechanism of Hb–Cu_3_(PO_4_)_2_ NFs, terephthalic acid (TA) was used as a fluorescence probe for the tracking of ·OH because it can capture ·OH and generate a fluorescent product, 2-hydroxyterephthalic acid (HTA), at 425 nm. As shown in Appendix A, the control groups, TA, TA with H_2_O_2_ and TA with Hb–Cu_3_(PO_4_)_2_ NFs, did not show significant intensity for HTA. Only in the presence of Hb–Cu_3_(PO_4_)_2_ NFs and H_2_O_2_ could fluorescence be found. This result supports the hypothesis that Hb–Cu_3_(PO_4_)_2_ NFs can catalyze H_2_O_2_ to generate ·OH, thus demonstrating the peroxidase-like activities of Hb–Cu_3_(PO_4_)_2_ NFs. Therefore, eight systems were established to prove that Hb–Cu_3_(PO_4_)_2_ NFs are peroxidases. Figure 2 clearly shows that fluorescence intensity increased with the presence of Hb–Cu_3_(PO_4_)_2_ NFs and H_2_O_2_. No fluorescence intensity was observed in the absence of H_2_O_2_ or Hb–Cu_3_(PO_4_)_2_ NFs. In addition, an equal amount of Hb or copper ion (1 mg/mL) in the presence of both TH and H_2_O_2_ showed weak fluorescence (Appendix A). These results indicate that Hb–Cu_3_(PO_4_)_2_ NFs can catalytically activate H_2_O_2_ decomposition to generate·OH radicals and lead to more efficient oxidation of thiamine.

According to a previous report [24], Hb–Cu_3_(PO_4_)_2_ NFs showed peroxidase activity under acidic pH conditions. However, the above fluorescence experiment results indicate that Hb–Cu_3_(PO_4_)_2_ NFs were highly active even at pH 10. Therefore, the effect of pH on the enzyme activity of Hb–Cu_3_(PO_4_)_2_ NFs was throughout measured by varying the pH and keeping the OPD and/or H_2_O_2_ concentration constant. As shown in Appendix A, the peroxidase-like activity of Hb–Cu_3_(PO_4_)_2_ NFs was present at pH 3 to 11. Interestingly, as shown in Appendix A, it was found that Hb–Cu_3_(PO_4_)_2_ NFs could catalyze oxidase reactions with OPD in the absence of H_2_O_2_ and showed oxidase-like activity at pH 12 and 13. In contrast, no obvious absorbance was found for other pH solutions The results indicate that the Hb–Cu_3_(PO_4_)_2_ NFs actualized dual enzyme-like activity. In Appendix A, it can be observed that the OPD oxidation rate catalyzed by the Hb–Cu_3_(PO_4_)_2_ NFs was dependent on the concentration of nanoflowers and period of time at pH 10. The absorbance values were increased with an increasing amount of NFs and length of time. According to a method described previously [26,27,28,29,30], the mechanism of peroxidase-like catalytic activities of Hb–Cu_3_(PO_4_)_2_ NFs can be further investigated using steady-state kinetic assays at pH 10. As shown in Appendix A, a typical Michaelis–Menten curve was obtained for Hb–Cu_3_(PO_4_)_2_ NFs in the presence of H_2_O_2_ and OPD, respectively. The Michaelis–Menten constant (K_m_) and the maximum initial velocity (V_max_) were acquired from the Michaelis–Menten curve. The K_m_ of Hb–Cu_3_(PO_4_)_2_ NFs with OPD and H_2_O_2_ as substrates was 0.02 mM and 1.8 mM, respectively. The V_max_ of Hb–Cu_3_(PO_4_)_2_ NFs with OPD and H_2_O_2_ as substrates was 1.96 × 10^−8^ M × S^−1^ and 1.38 × 10^−8^ M × S^−1^, respectively. Overall, the enhancement in the enzymatic activity of Hb–Cu_3_(PO_4_)_2_ NFs can be ascribed to the possible stabilization of the NF-like structure of Hb through high surface area and confinement, resulting in higher accessibility of the substrate to the active sites. Thus, thiamine could be oxidized to yield fluorescent thiochrome in a basic solution. As shown in Figure 3, the fluorescence values of thiochrome were measured by a fluorescence spectrophotometer with a maximum emission of 441 nm and an excitation wavelength of 370 nm.

### 3.3. Condition Optimization

A series of experiments were conducted to establish the optimum analytical conditions for the oxidation of thiamine by H_2_O_2_ with Hb–Cu_3_(PO_4_)_2_ NFs as catalyst. The parameters investigated were reaction time; the concentrations of pH, H_2_O_2_ and Hb–Cu_3_(PO_4_)_2_ NFs; and the fluorescence sensitizer, Tween 80.

#### 3.3.1. Effect of pH and Reaction Time

The solution pH was an important factor which influenced the peroxidase-like activity of the Hb–Cu_3_(PO_4_)_2_ NFs. As shown in Figure 4a, when using different phosphate buffers ranging from pH 8 to 13, the peak of the maximum fluorescence intensity was obtained with a phosphate buffer of pH 10. The fluorescence intensity of thiochrome rapidly increased with reaction time, initially from 0 to 300 s, and then the values slowly reached a plateau, indicating that the H_2_O_2_ was completely consumed by oxidation of the substrate TH. Thus, 300 s was chosen as the optimal reaction time for a phosphate buffer of pH 10.

#### 3.3.2. Effect of H_2_O_2_ Concentration

To achieve a maximum oxidation rate of the thiamine, optimization of hydrogen peroxide concentration was necessary. As shown in Figure 4b, the fluorescence intensity increased quickly with the concentration of H_2_O_2_ up to 100 mM and then decreased. Therefore, 100 mM H_2_O_2_ was selected as the optimum concentration.

#### 3.3.3. Effect of Hb–Cu_3_(PO_4_)_2_ NF Concentration

In order to achieve the optimal oxidation rate of the thiamine catalyzed by Hb–Cu_3_(PO_4_)_2_ NFs, the effect of Hb–Cu_3_(PO_4_)_2_ NF concentrations (0–4 mg/mL) on the fluorescence of thiochrome was investigated. As shown in Figure 4c, the fluorescence intensity first increased and then slowly decreased with an increase in Hb–Cu_3_(PO_4_)_2_ NF concentration. The maximum fluorescence intensity in the system was obtained at 2 mg/mL of Hb–Cu_3_(PO_4_)_2_ NF solution. Thus, 2 mg/mL was selected for the Hb–Cu_3_(PO_4_)_2_ NFs as the optimal concentration.

#### 3.3.4. Effect of Tween 80 Concentration

Tween 80, a hydrophilic nonionic surfactant, had been utilized in the drug loading and the biosensor fields for its hydrophilicity and low toxicity [31,32,33]. Previous reports have shown that Tween 80 can enhance the fluorescence of a system. For example, it has been used in fluorescence detections of nicotinamide [34] and nifanib [35]. In this work, Tween 80 was added in the fluorescence determination for thiamine. The effect of the amount of Tween 80 (0.1–0.5%) was investigated with the other parameters kept constant. As shown in Figure 4d, the fluorescence intensity first increased and then slowly decreased along with an increase in the amount of Tween 80. The maximum fluorescence intensity in the system was obtained at a Tween 80 solution of 0.4%. Therefore, a Tween 80 solution of 0.4% was chosen as the optimal concentration.

### 3.4. Calibration Curve for Thiamine Detection

Under the optimized reaction conditions, the relationship between the fluorescence intensity and thiamine concentration catalyzed by Hb–Cu_3_(PO_4_)_2_ NFs was investigated. As shown in Figure 5, the fluorescence intensity increased with increasing concentration of thiamine. The linear regression equation was F = 19.247 + 2.452 × 10^8^ × C_thiamine_ with a correlation coefficient of 0.9984. There was a good linear correlation between the fluorescence intensity and the thiamine concentration in the range of 5 × 10^−8^ to 5 × 10^−5^ M. The lower limit of detection of Hb–Cu_3_(PO_4_)_2_ NFs for thiamine was found to be 4.8 × 10^−8^ M, which is similar to the HRP-based fluorescent method for the detection of thiamine [36]. It is almost believable that its ability to detect thiamine is similar to the HRP-based fluorescent method. Compared with HRP, nanozymes have the advantages of being highly stable against denaturing, low-cost, easy to store and suitable for treatment. The mimic enzyme activity of Hb–Cu_3_(PO_4_)_2_ NFs was firstly utilized in fluorometric sensing of thiamine under basic conditions. Compared with other systems for the determination of thiamine, as shown in Table 1, the proposed method based on Hb–Cu_3_(PO_4_)_2_ NFs is on a par with other fluorescence methods. Additionally, the synthesis of the Hb–Cu_3_(PO_4_)_2_ NFs was convenient and simple. Therefore, the method should be a potential candidate for fluorescence sensors of thiamine.

### 3.5. Determination of Thiamine in Functional Food Tablet Samples

In order to evaluate the feasibility of the proposed method based on Hb–Cu_3_(PO_4_)_2_ NFs, the concentrations of thiamine in multivitamin functional food tablets samples were analyzed under the optimal conditions. As shown in Table 2, the recovery level of the developed approach was between 103.20 and 115.20%. The result suggests that the developed method is accurate and reliable for thiamine detection in real samples.

### 3.6. Interference Study

The selectivity of the proposed method was examined by comparing the fluorescence change of the solution with thiamine (0.3 mg/mL) with other coexistence substrates (30 mg/mL) including K^+^, Na^+^, Cl^-^, HCO^3-^, glucose, starch, VB_2_, VB_7_, VB_9_ and Vc. These control compounds are usually found in foods. As shown in Figure 6, the fluorescence of the solution changed only when thiamine was added to the mixture. The addition of the other molecules, in contrast, had no obvious effect on the fluorescence of the solutions. Thus, the proposed fluorometric method displayed a high selectivity for the determination of thiamine.

## 4. Conclusions

In summary, a new fluorometric enhancement method for the detection of thiamine has been demonstrated. The peroxidase-like activity of Hb–Cu_3_(PO_4_)_2_ NFs was first found in a basic solution. Under the optimal conditions, H_2_O_2_ could be decomposed into·OH radicals by Hb–Cu_3_(PO_4_)_2_ NFs that oxidize thiamine rapidly and efficiently. A linear correlation was established between fluorescence intensity and the concentration of thiamine from 0.05–50 μM with a detection limit of 48 nM. Future research may focus on attempts to explore the new protein–inorganic hybrid nanoflowers or analytes in the detecting systems.

## Figures and Tables

**Figure 1 sensors-20-06359-f001:**
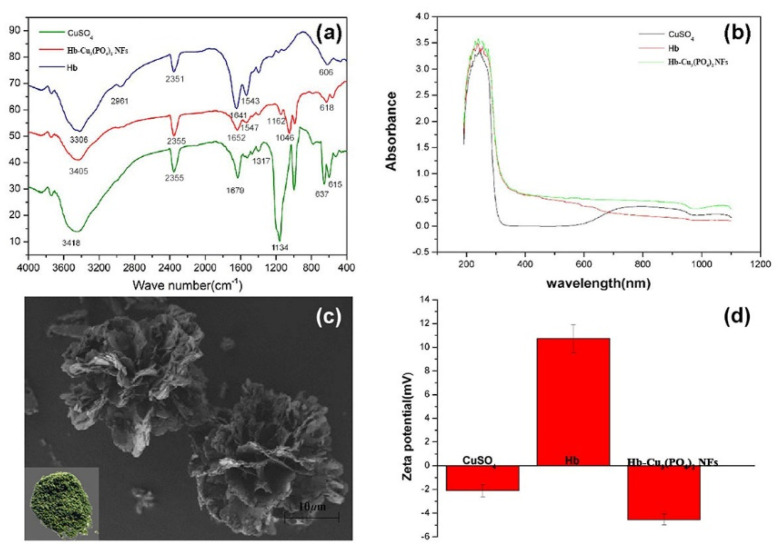
(**a**) IR spectra of CuSO_4_ (green line), Hb (blue line) and Hb–Cu_3_(PO_4_)_2_ NFs (red line); (**b**) UV–vis absorptions of CuSO_4_ (blue line), Hb (red line) and Hb–Cu_3_(PO_4)2_ NFs (green line); (**c**) SEM image of Hb–Cu_3_(PO_4_)_2_ NFs; (**d**) Zeta potentials of CuSO_4_, Hb and Hb–Cu_3_(PO_4_)_2_ NFs.

**Figure 2 sensors-20-06359-f002:**
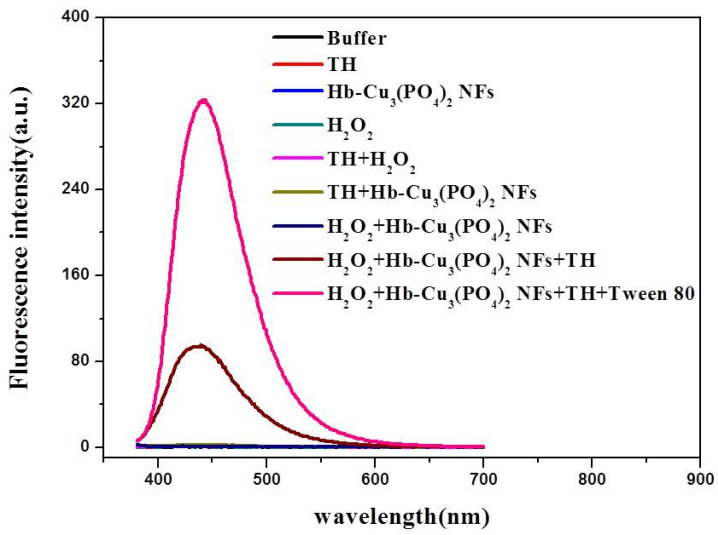
Fluorescence spectra of Na_2_HPO_4_-NaOH (pH 10) (black line), TH (red line), Hb–Cu_3_(PO_4_)_2_ NFs (light blue line), H_2_O_2_(dark green line), TH + H_2_O_2_ (purple line), TH + Hb–Cu_3_(PO_4_)_2_ NFs (light green line), Hb–Cu_3_(PO_4_)_2_ NFs + H_2_O_2_ (dark blue line), TH + H_2_O_2_ + Hb–Cu_3_(PO_4_)_2_ NFs (brown line), TH + H_2_O_2_ + Hb–Cu_3_(PO_4_)_2_ NFs + Tween 80 (pink line).

**Figure 3 sensors-20-06359-f003:**
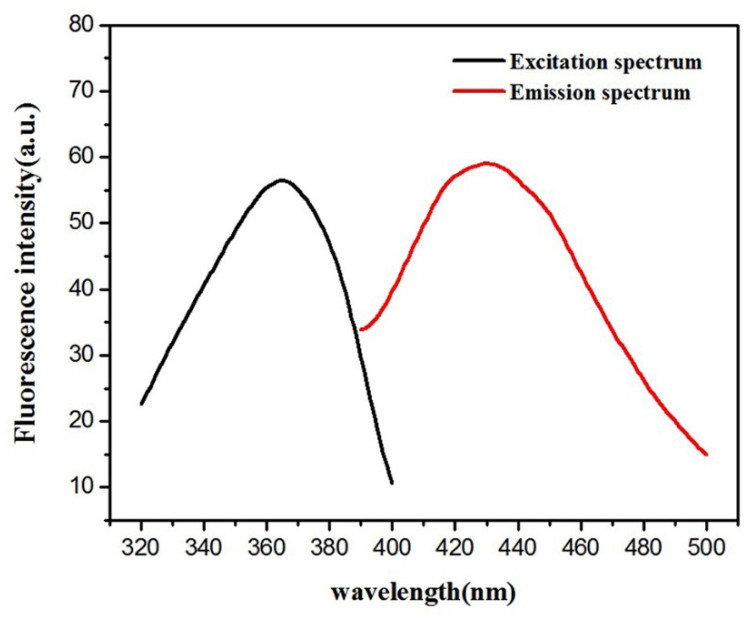
The excitation and emission spectra for thiochrome.

**Figure 4 sensors-20-06359-f004:**
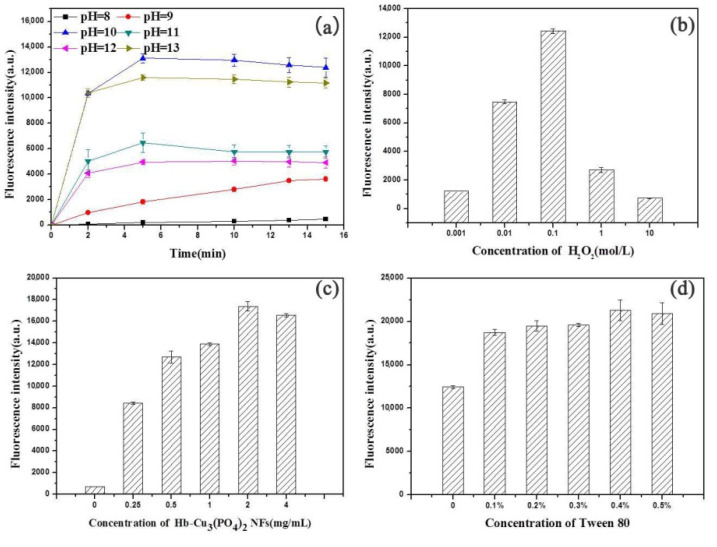
The optimization of experimental conditions: (**a**) reaction time and pH, (**b**) H_2_O_2_ concentration, (**c**) Hb–Cu_3_(PO_4_)_2_ NF concentration and (**d**) the amount of Tween 80. Conditions: (**a**) H_2_O_2_, 1 × 10^−3^ mol/L; Hb–Cu_3_(PO_4_)_2_ NFs, 2 mg/mL; thiamine, 1 × 10^−3^ mol/L. (**b**) Hb–Cu_3_(PO_4_)_2_ NFs, 2 mg/mL; thiamine, 1 × 10^−3^ mol/L; pH, 10; incubation time, 5 min. (**c**) H_2_O_2_, 1 × 10^−3^ mol/L; thiamine, 1 × 10^−3^ mol/L; pH, 10; incubation time, 5 min. (**d**) H_2_O_2_, 1 × 10^−3^ mol/L; Hb–Cu_3_(PO_4_)_2_ NFs, 2 mg/mL; thiamine, 1 × 10^−3^ mol/L; pH, 10; incubation time, 5 min.

**Figure 5 sensors-20-06359-f005:**
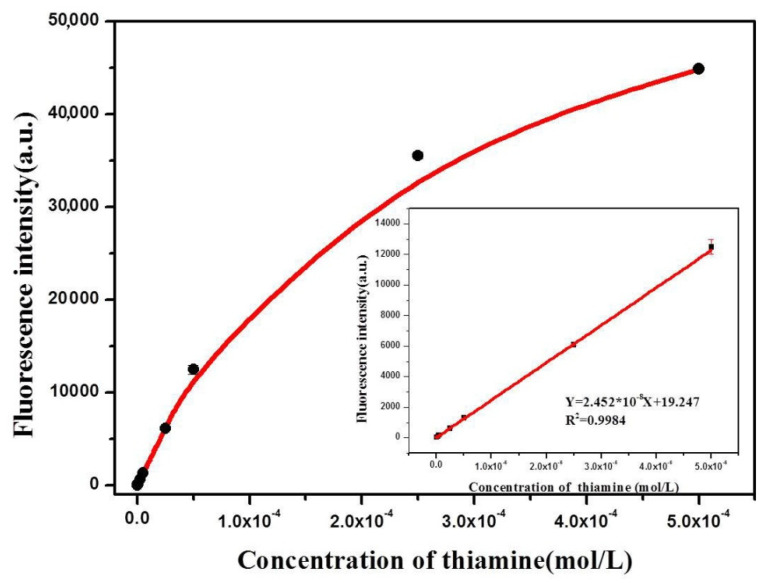
Fluorescence intensity change of the solutions containing Hb–Cu_3_(PO_4_)_2_ NFs upon the addition of different concentrations of thiamine (5 × 10^−9^ to 5 × 10^−4^ mol/L). The inset shows the linear calibration plot for TH. Error bar: RSD (*n* = 3).

**Figure 6 sensors-20-06359-f006:**
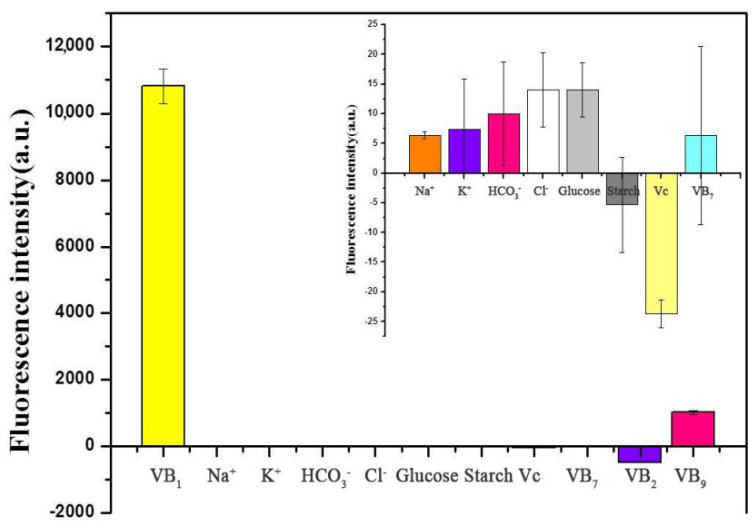
Interference study for the determination of thiamine. Concentrations of Na^+^, K^+^, HCO_3_^-^, Cl^-^, glucose, starch, Vc, VB_7_, VB_2_ and VB_9_: 30 μg/mL. Inset: the amplified fluorescence intensity of Na^+^, K^+^, HCO_3_^-^, Cl^-^, glucose, starch, Vc and VB_7_.

**Table 1 sensors-20-06359-t001:** Comparison of the detection of thiamine in different fluorescent systems.

System	Linear Range	Detection Limit	Reference
TBP/IMS/FRET	5–240 nM	2 nmol/g	[36]
e-PNPs/ESIPT	0.1–25 μM	2.6 nM	[7]
O-phen/Zn^2+^	0.84–80.0 μM	0.25 μM	[37]
HKUST-1	4–700 μM	1 μM	[38]
C-dots/Cu^2+^	10–50 μM	0.28 nM	[39]
HRP	0.08–49.90 μM	0.04 μM	[2]
Cu^2+^	0.89–17.85 μM	0.50 μM	[40]
Hb–Cu_3_(PO_4_)_2_ NFs	0.05–50 μM	0.048 μM	This work

TBP: thiamine periplasmic binding protein; IMS: immunomagnetic separation; FRET: fluorescence resonance energy transfer; e-PNPs: exhibiting polymer nanoparticles; ESIPT: excited-state intramolecular proton transfer; O-phen: o-phenanthroline; HKUST-1: peroxidase-like activity of copper-based MOFs; C-dots: carbon dots; HRP: horseradish peroxidase.

**Table 2 sensors-20-06359-t002:** The recovery of thiamine in tablets by standard addition.

Sample	Added (μM)	Detected (μM)	Recovery (%)
1	25	25.8 ± 4.35	103.2
2	0.50	0.57 ± 0.03	115.2
3	0.25	0.27 ± 0.19	107.2

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
