# Peer review of "Fluorometric Detection of Thiamine Based on Hemoglobin–Cu3(PO4)2 Nanoflowers (NFs) with Peroxidase Mimetic Activity"

_sensors, 2020, doi:10.3390/s20216359_

Round 1

Reviewer 1 Report

The authors describe the use of a hemoglobin-copper complex in an alkaline environment as a component of a fluorometric method for thiamine detection. I consider their work interesting and properly conducted; I suggest to publish it after the authors will have answered to some comments

  • In the Condition Optimization section (3.3) only the changing parameter is reported, not the other parameters kept fix
  • There is no clear trend in the effect of the pH discussed in 3.3.1 and Figure.4a. Could the authors explain why at pH=13 they observed a similar behavior to pH =10?
  • Line 249: the use of the passive voice for the fluorescent intensity sounds strange, make it active: " the fluorescence intensity firstly increased and then slowly decreased..."
  • 3.3.4: taking into account the error bars on Figure.4d the fluorescence intensity is quite similar for a concentration of Tween 80 between 0.1 and 0.5
  • 3.5 what do the authors mean with "Recovery /%"? 
  • I did not have access to the Supporting Information file, not able to do a complete review
  • The introduction and Materials and Methods sections are well written, while the Results and Discussion must be checked carefully, some phrases are not clear and verbs conjugation is not coherent
  • Minor spells and typos:

line 89: "...modified[24]." should be "modification [24]."

line 107: Na2HPO4-NaOH numbers should be subscripts

line 125 and 142 "...buffer(pH..." a space is missing

line 237 "plateau" instead of "platform"

...

Reviewer 2 Report

This paper is on experimental study of nanostructured organic-inorganic materials for Vitamin B1 detection. The data support conclusion. It is acceptable for publishing upon some minor changes.

Here is the list of suggested changes:

1. A typo is identified in the Introduction part on line 34, "fluorescence[7,16,17] ,electrochemistry[18] " should be corrected as "fluorescence[7,16,17], electrochemistry[18] " (A space is needed before the last word.).

2. On line 44, "highly stability" should be changed into "high stability".

3. In Materials and Experimental Section, line 68, "KCl,NaHCO3" should be "KCl, NaHCO3". (A space between two formulae is needed.)

4. In subsection 2.3, line 89, "the literature method with little modified" is not a good expression. It is suggested to change.
Either, "the literature reported method" or "the literature method without modification".

5. On line 144, " in 37" should be "at 37".

6. On line 145, "with 5 min" is better to change into "for 5 min".

7. On line 195, "According to previous reports [24]" sould be "According to the previously reported results in [24]" or "According to a previous report [24]".

8. On line 204, "From 3 to 11" is confusion here. Maybe "In pH values range from 3 to 11" is better.

9. In the cited reference [37], either volume number or page number is missing.

10. On line 310, "Interference Study: the" sould be "Interference Study: The".

Round 2

Reviewer 1 Report

I thank the authors for their answers and manuscript new version.

I find it suitable for publication after a final check for minor typos and writing errors.

Author Response

Response to Reviewer 1 Comments:

Point 1: I find it suitable for publication after a final check for minor typos and writing errors.

Response 1:Thank you for your nice advice. We have throughtly checked our manuscript and revised the tpos and writing errors. Thanks again.